# Pranlukast treatment and the use of respiratory support in infants with respiratory syncytial virus infection

Jun Kubota[1]*, Sho Takahashi[2], Takayuki Suzuki[1,3], Akira Ito[1,3], Naoe Akiyama[1,4], Noriko Takahata[1]

1 Department of Pediatrics, The Jikei University School of Medicine, Tokyo, Japan, 2 Clinical Research Support Center, The Jikei University School of Medicine, Tokyo, Japan, 3 Department of Pediatrics, Atsugi City Hospital, Kanagawa, Japan, 4 Department of Pediatrics, Fuji City General Hospital, Shizuoka, Japan

* junkubota13@gmail.com

## Abstract

### Background

In infants, respiratory syncytial virus (RSV) infection occasionally causes severe symptoms requiring respiratory support; however, supportive care is the primary treatment. This study compared the use of respiratory support among infants with RSV infection treated with or without pranlukast.

### Methods

This retrospective cohort study included infants aged <10 months with RSV infection who were admitted to three secondary level hospitals in Japan between 2012 and 2019. The infants were divided into two groups depending on whether they were treated with pranlukast. The primary outcome was the receiving respiratory support (high-flow nasal cannula, nasal continuous positive airway pressure, or ventilator). The secondary outcomes were the length of hospital stay, and the Global Respiratory Severity Score (GRSS) on starting respiratory support or at the time of the worst signs during hospitalization. We performed a propensity score-matched analysis.

### Results

A total of 492 infants, including 147 propensity score-matched pairs, were included in the analysis. The use of respiratory support was significantly lower in infants treated with pranlukast (3.4% [5/147]) than those treated without pranlukast (11.6% [17/147]; $P = 0.01$). In the propensity score-matched analysis, pranlukast use was associated with a significantly lower chance of needing respiratory support (odds ratio: 0.27, 95% confidence interval: 0.08–0.79; $P = 0.01$); however, the length of hospital stay (median: 4 days) and the GRSS (median: 2.804 and 2.869 for infants treated with and without pranlukast, respectively) did not differ significantly between propensity score-matched pairs.

**Data Availability Statement:** All relevant data are within the paper and its Supporting Information files.

**Funding:** The authors received no specific funding for this work.

**Competing interests:** The authors have declared that no competing interests exist.

## Conclusions

Pranlukast use was associated with a reduced likelihood of requiring respiratory support in infants aged <10 months with RSV infection.

## Introduction

Respiratory syncytial virus (RSV) is a common cause of acute respiratory infection in children [1, 2]. RSV infection has a broad range of severity from mild to severe disease requiring ventilator support [3]. RSV is one of the most common causes of pediatric hospitalization, and more than 80% of infants hospitalized with RSV infection are healthy without any underlying conditions [4]. Supportive care is the only primary treatment because there is no specific treatment for RSV [5–7]. One of the most important strategies for treating patients with RSV infection is to decide whether the patients need respiratory support [8, 9]. From a health economics viewpoint, this has important implications. In children with bronchiolitis aged less than 2 years, the use of mechanical ventilation significantly increases hospital costs; however, the overall hospitalization rate for bronchiolitis decreased between 2000 and 2016 [10, 11]. Cysteinyl leukotriene receptor antagonists (LTRAs), which have anti-inflammatory effects and effects on bronchoconstriction and bronchial hyperresponsiveness [12, 13], may be useful for treating viral infections [13–15]. Cysteinyl leukotrienes are increased during RSV infection [7, 16–18]. Mixed results were obtained in studies on the use of the LRTA, montelukast, for the treatment of acute viral bronchiolitis. Some studies found that montelukast may improve the symptoms of acute viral bronchiolitis [19, 20], while others found that montelukast did not improve the acute symptoms of RSV infection [21–23]. However, from the viewpoint of health care costs, it is important to consider whether LTRAs reduce the number of patients with RSV infection who need respiratory support, not whether they improve symptoms. If LTRAs reduce the need for respiratory support, they also reduce the physical and psychological burdens on patients and their families caused by using respiratory support.

This study aimed to evaluate the effect of LTRAs on the likelihood of requiring respiratory support (high-flow nasal cannula, nasal continuous positive airway pressure, or ventilator) in infants with RSV infection.

## Methods

### Study population

We conducted a retrospective cohort study at the Jikei University Katsushika Medical Center in Tokyo, Atsugi City Hospital in Kanagawa, and Fuji City General Hospital in Shizuoka, Japan. All three hospitals are secondary level facilities and are in charge of pediatric emergency treatment of children in separate catchment areas. Therefore, each hospital was the first-visited hospital of children living in the area who required emergency hospital care. We included infants aged <10 months who were admitted to the Departments of Pediatrics at the three hospitals for the treatment of RSV infection between January 1, 2012, and December 31, 2019. Only infants aged <10 months were included because we assessed the disease severity of the included infants using the Global Respiratory Severity Score (GRSS), which was developed as a specific scoring system to measure the overall severity over the course of RSV infection in infants aged <10 months [24]. RSV infection was diagnosed using commercial rapid antigen-based tests. The tests differed across facilities, were revised over time, and included Alere

BinaxNOW RSV rapid test (Abbott Diagnostics, Abbott Park, IL, USA), ALSONIC RSV (Alfresa Pharma Corp., Osaka, Japan), Check RSV (Meiji Seika Pharma Co., Ltd., Tokyo, Japan), ImmunoAce RSV Neo (Tauns Laboratories, Inc., Shizuoka, Japan), and RapidTesta RSV-Adeno NEXT (Sekisui Chemical Co., Ltd., Tokyo, Japan). Infants with any of the following conditions were excluded based on the exclusion criteria of the GRSS [24]: a gestational age at birth <36 weeks; hospitalization for apnea only; high-risk conditions, such as chronic aspiration, congenital heart disease, immunosuppression, malignancy, and neurological conditions; and indications for palivizumab prophylaxis. In addition, we excluded infants with a history of admission in the neonatal intensive care unit, asthma or previous wheezing, treatment with steroids before or during hospitalization, or a lack of the basic information needed to calculate the GRSS.

Starting respiratory support (with a high-flow nasal cannula, nasal continuous positive airway pressure, or ventilator) was dependent on the pediatrician's judgment based on clinical signs such as retractions, tachypnea, the presence of wheeze/rales/rhonchi, or respiratory acidemia on venous blood gas analysis.

Across Japan, the number of RSV infections peaks in the winter season. The overall reported incidence increased by 40% during the study period (https://www.niid.go.jp/niid/ja/ydata/10071-report-jb2019.html primarily because of the increased use of rapid antigen test kits and expanded insurance coverage for the use of rapid antigen tests in Japan.

## Study design

We reviewed the medical records of each patient. The primary outcome was the use of respiratory support (high-flow nasal cannula, nasal continuous positive airway pressure, or ventilator). The secondary outcomes were the length of hospital stay without administrative and social factors (without supplemental oxygen for 10 hours, minimal or no chest recession, and adequate feeding) [25] and the GRSS on starting respiratory support or at the time of the most severe point of the illness. During hospitalization, the GRSS was calculated each day by entering the following 10 parameters: age (months), oxygen saturation (%), respiratory rate (breaths/minute), general appearance, presence of wheeze, rales/rhonchi, retractions, cyanosis, lethargy, and poor air movement in the online calculator (available at: https://rprc.urmc.rochester.edu/app/AsPIRES/RSV-GRSS/) [24]. We calculated this score using the worst signs recorded each day.

We divided the infants into two groups depending on whether they were treated with an LTRA. We used pranlukast (7 mg/kg twice daily) rather than montelukast as the LTRA, based on insurance coverage in Japan. If infants started pranlukast on the same day as starting respiratory support, they were assigned to the no pranlukast group. The sick days were calculated using the day of onset of the presenting signs (rhinorrhea, cough, wheezing, fever, and lethargy) as the first day.

## Statistical analysis

Continuous variables are expressed as the median and interquartile range (IQR), and categorical variables are expressed as frequencies. Statistical comparisons between the two groups were performed using the Mann–Whitney U-test for continuous variables and the chi-square or Fisher's exact test for categorical variables. Two-sided P values <0.05 were considered statistically significant. Missing data were treated as missing, with no imputation of missing values.

A propensity score-matched analysis was performed, and the average treatment effect was estimated using the inverse probability of treatment weighting. First, propensity score matching was performed in a 1:1 ratio, matching patients treated with pranlukast with infants not

treated with pranlukast using the nearest-neighbor method within a caliper distance of less than 20% of a standard deviation for the propensity score [26]. Propensity scores were calculated using multivariable logistic regression models to establish each patient's probability of receiving pranlukast according to baseline characteristics (sex, age, gestational age at birth, sick days on admission, the GRSS on admission, oxygen support, antibiotics, bronchodilator inhalation, and hospitals). The balance between the two groups was checked based on absolute standardized differences [26]. If the absolute standardized difference was less than 0.1, it was considered a meaningful balance. The baseline characteristics and outcomes were compared using the propensity score-matched cohort. Second, to check the robustness of the study findings, the average treatment effect was estimated using the inverse probability of treatment weighting [26, 27]. To evaluate the consistency of starting respiratory support, the infants were divided into two groups depending on whether they received respiratory support. The GRSS on admission and the GRSS on starting respiratory support or at the time of the most severe point of the illness of children in each group were compared. Furthermore, within each group, patients were compared according to whether they were treated with pranlukast. Multivariable logistic regression was used to estimate odds ratios (ORs) and 95% confidence intervals (CIs).

All analyses were performed using Stata version 15.1 (StataCorp LP, College Station, TX, USA) software package. The statistical code used for running propensity score matching and the balance check based on standardized differences is available at: https://ideas.repec.org/c/boc/bocode/s432001.html, http://personalpages.manchester.ac.uk/staff/mark.lunt). The data were analyzed in June and July 2021.

### Ethical approval and informed consent

This study was conducted in accordance with the ethical principles of the Declaration of Helsinki and with the ethical guidelines for epidemiological studies issued by the Ministry of Health, Labour and Welfare, Japan. This study was approved by the Institutional Review Boards of the Jikei University School of Medicine (33-026(10636)), Atsugi City Hospital (R3-06), and Fuji City General Hospital (259). The requirement for obtaining informed consent from the patients' guardians was waived because the data were obtained retrospectively.

## Results

### Patient characteristics

Fig 1 shows a flowchart of the study design. A total of 814 infants with RSV infection aged <10 months were admitted to three hospitals during the study period. Of these infants, 322 (39.6%) did not meet the inclusion criteria and were excluded from the study. Of the remaining 492 infants (median [IQR] age: 2.8 [1.6 to 5.4] months, 263 [53.5%] males), 198 (40.2%) were treated with pranlukast. Pranlukast was started on admission, before admission, and after admission in 78.8% (156/198), 12.1% (24/198), and 9.1% (18/198) of the infants treated with pranlukast, respectively.

Propensity scores were calculated based on sex, age, gestational age at birth, days since onset on admission, GRSS on admission, oxygen support, antibiotic use, bronchodilator inhalation, and hospital.

The distribution of the propensity score between infants with and without pranlukast overlapped between the two groups (Fig 2). After propensity score matching, 147 patients were assigned to each group. The balance was satisfactory because the absolute standard differences of all variables included in the matching process were less than 0.1 (Fig 3). The baseline characteristics of each group before and after propensity score matching are shown in Table 1.

The GRSS on starting respiratory support or at the time of the most severe point of the illness was significantly higher in the group that received respiratory support than in the group

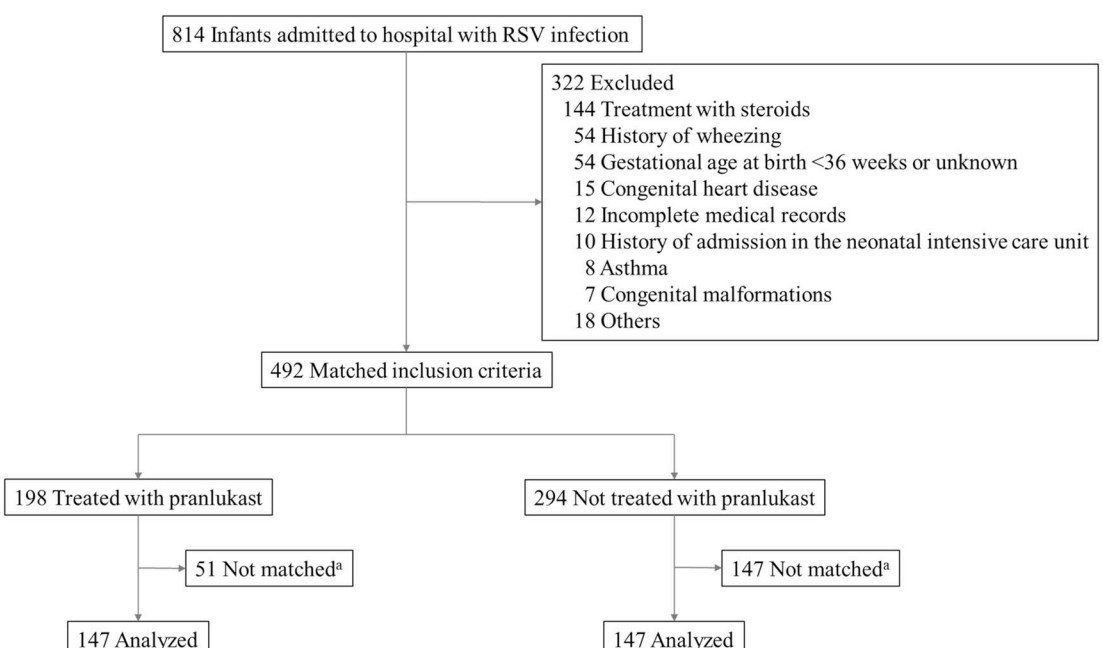

**Fig 1. Study flow chart.** [a] Infants treated with pranlukast were propensity score-matched with infants not treated with pranlukast in a 1:1 ratio using nearest-neighbor matching.

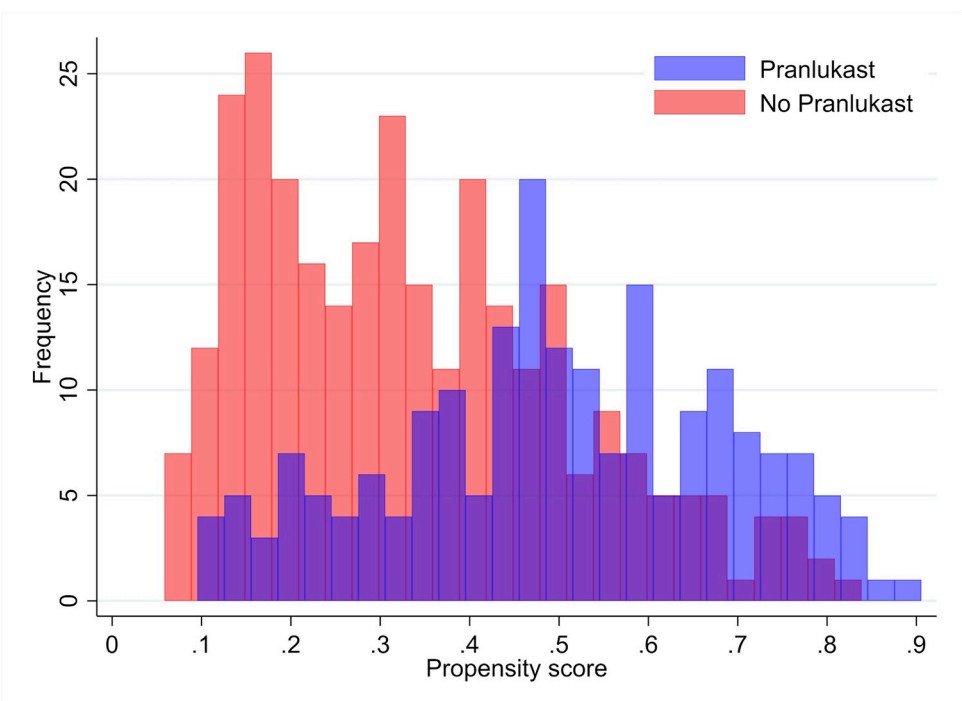

**Fig 2. Histogram of propensity score distribution of infants treated with (blue bars) and without (red bars) pranlukast.**

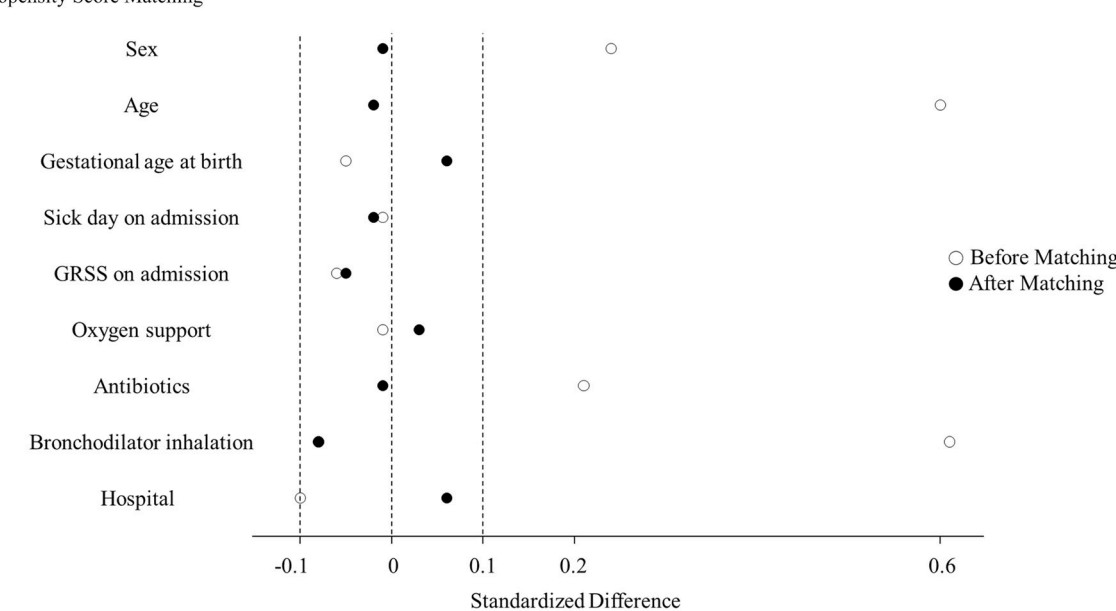

**Fig 3. Covariate balance for propensity score matching of infants treated with and without pranlukast.**

that did not receive respiratory support (Tables 2 and 3). However, the GRSS did not differ significantly according to whether the patient was treated with pranlukast in regardless of whether respiratory support or not (Tables 2 and 3). These results were consistent, regardless of propensity score matching. None of the infants included in the analysis experienced any adverse events.

## Primary outcome

After propensity score matching, the incidence rate of receiving respiratory support in infants treated with pranlukast was significantly lower than that of the infants who were not treated with pranlukast (3.4% [5/147] and 11.6% [17/147], respectively; $P = 0.01$) (Table 1).

Pranlukast was associated with a lower rate of respiratory support (OR: 0.27, 95% CI: 0.08–0.79; $P = 0.01$) (Table 4). Similar results were observed when the data were analyzed using inverse probability of treatment weighting for the average treatment effect (OR: 0.22, 95% CI: 0.07–0.63; $P = 0.01$) (Table 4). After the inverse probability of treatment weighting, the standard differences of all variables included were <0.1, indicating that the balance was satisfactory (Fig 4).

## Secondary outcomes

After propensity score matching, the length of hospital stay and the GRSS on starting respiratory support or at the time of the worst signs were not significantly different between the groups (4 days, $P = 0.73$; 2.804 and 2.869, $P = 0.96$, respectively) (Table 1).

## Discussion

Pranlukast reduced the rate of respiratory support in infants with RSV infection. In this study, the result of the average treatment effect using inverse probability of treatment weighting showed that pranlukast significantly reduced the likelihood of requiring respiratory support.

**Table 1. Baseline characteristics of the study patients before and after propensity score matching.**

| | Before propensity score matching | | | | After propensity score matching | | | |
|---|---|---|---|---|---|---|---|---|
| | Pranlukast | No pranlukast | P value | Standardized difference | Pranlukast | No pranlukast | P value | Standardized difference |
| | (n = 198) | (n = 294) | | | (n = 147) | (n = 147) | | |
| Sex | | | 0.01 | 0.24 | | | 0.91 | −0.01 |
| Male, n (%) | 120 (60.6) | 143 (48.6) | | | 77 (52.4) | 78 (53.1) | | |
| Female, n (%) | 78 (39.4) | 151 (51.4) | | | 70 (47.6) | 69 (46.9) | | |
| Age, median (IQR), months | 3.8 (2.4–6.5) | 2.2 (1.3–4.0) | <0.001 | 0.60 | 3.2 (2.0–5.3) | 2.9 (1.5–6.0) | 0.40 | −0.02 |
| Gestational age at birth, median (IQR), weeks | 38.9 (38.0–39.9) | 39.0 (38.0–39.9) | 0.59 | −0.05 | 39.0 (38.0–40.0) | 38.9 (38.0–39.9) | 0.51 | 0.06 |
| Sick day, median (IQR), days | 4 (3–5) | 4 (3–5) | 0.89 | −0.01 | 4 (3–5) | 4 (3–6) | 0.81 | −0.02 |
| GRSS on admission, median (IQR) | 2.483 (1.628–3.577) | 2.572 (1.237–3.892) | 0.93 | −0.06 | 2.490 (1.665–3.723) | 2.614 (1.390–3.892) | 0.90 | −0.05 |
| Respiratory support, n (%) | 5 (2.5) | 51 (17.3) | <0.001 | −0.51 | 5 (3.4) | 17 (11.6) | 0.01 | −0.31 |
| High-flow nasal cannula, n | 1 | 19 | | | 1 | 6 | | |
| Nasal continuous positive airway pressure, n | 1 | 18 | | | 1 | 5 | | |
| Ventilator, n | 3 | 14 | | | 3 | 6 | | |
| GRSS on starting respiratory support or at the time of the worst signs, median (IQR) | 2.789 (1.972–3.847) | 2.982 (1.863–4.099) | 0.45 | −0.12 | 2.804 (1.977–4.015) | 2.869 (1.727–3.999) | 0.96 | −0.06 |
| Oxygen support, n (%) | 115 (58.1) | 172 (58.5) | 0.93 | −0.01 | 88 (59.9) | 86 (58.5) | 0.81 | 0.03 |
| Antibiotics, n (%) | 79 (39.9) | 88 (29.9) | 0.02 | 0.21 | 51 (34.7) | 52 (35.4) | 0.90 | −0.01 |
| Bronchodilator inhalation, n (%) | 159 (80.3) | 155 (52.7) | <0.001 | 0.61 | 109 (74.2) | 114 (77.6) | 0.50 | −0.08 |
| Length of the hospital stay, median (IQR), days | 4 (3–6) | 5 (3–6) | 0.84 | 0.02 | 4 (3–6) | 4 (3–6) | 0.73 | 0.06 |

GRSS, Global Respiratory Severity Score; IQR, interquartile range

**Table 2. Global Respiratory Severity Score on starting respiratory support or at the time of the worst signs after propensity score matching.**

| | With respiratory support (n = 22) | No respiratory support (n = 272) | P value | Standardized difference |
|---|---|---|---|---|
| GRSS on admission, median (IQR) | 4.107 (3.038–5.585) | 2.483 (1.418–3.596) | <0.001 | −1.07 |
| GRSS on starting respiratory support or at the time of the worst signs, median (IQR) | 4.423 (3.654–5.585) | 2.723 (1.799–3.853) | <0.001 | −1.21 |
| | With respiratory support | | | |
| | With pranlukast (n = 5) | No pranlukast (n = 17) | | |
| GRSS on admission, median (IQR) | 3.200 (3.038–3.387) | 5.324 (3.174–5.784) | 0.11 | 1.00 |
| GRSS on starting respiratory support or at the time of the worst signs, median (IQR) | 4.229 (3.654–4.326) | 4.849 (3.665–5.784) | 0.46 | 0.45 |
| | No respiratory support | | | |
| | With pranlukast (n = 142) | No pranlukast (n = 130) | | |
| GRSS on admission, median (IQR) | 2.483 (1.654–3.723) | 2.483 (1.233–3.495) | 0.27 | −0.12 |
| GRSS on starting respiratory support or at the time of the worst signs, median (IQR) | 2.734 (1.972–3.847) | 2.723 (1.620–3.858) | 0.44 | −0.08 |

GRSS, Global Respiratory Severity Score; IQR, interquartile range

**Table 3. Global Respiratory Severity Score on starting respiratory support or at the time of the worst signs before propensity-score matching.**

| | With respiratory support (n = 56) | No respiratory support (n = 436) | P value | Standardized difference |
|---|---|---|---|---|
| GRSS on admission, median (IQR) | 4.557 | 2.341 | <0.001 | −1.23 |
| | (3.187–5.766) | (1.388–3.497) | | |
| GRSS on starting respiratory support or at the time of the worst signs, median (IQR) | 4.861 | 2.674 | <0.001 | −1.50 |
| | (3.660–5.779) | (1.779–3.726) | | |
| | With respiratory support | | | |
| | With pranlukast | No pranlukast | | |
| | (n = 5) | (n = 51) | | |
| GRSS on admission, median (IQR) | 3.200 | 4.787 | 0.049 | 0.99 |
| | (3.038–3.387) | (3.252–5.912) | | |
| GRSS on starting respiratory support or at the time of the worst signs, median (IQR) | 4.229 | 4.975 | 0.23 | 0.64 |
| | (3.654–4.326) | (3.665–5.912) | | |
| | No respiratory support | | | |
| | With pranlukast | No pranlukast | | |
| | (n = 193) | (n = 243) | | |
| GRSS on admission, median (IQR) | 2.455 | 2.218 | 0.02 | −0.20 |
| | (1.612–3.577) | (0.991–3.457) | | |
| GRSS on starting respiratory support or at the time of the worst signs, median (IQR) | 2.694 | 2.599 | 0.14 | −0.14 |
| | (1.955–3.829) | (1.620–3.666) | | |

GRSS, Global Respiratory Severity Score; IQR, interquartile range

This strategy, which differs from propensity score matching, has the advantage of including all the patients in the final analysis [27]. Therefore, the results were robust because the two different methods of analysis had similar results.

To our knowledge, this is the first study to evaluate the association between LTRA and the likelihood of requiring respiratory support in infants with RSV infection. The mechanism underlying the requirement of respiratory support by infants with severe RSV infection involves bronchiolar obstruction by increased mucus production and deposition of cellular debris [7, 15, 28] because RSV induces extensive inflammation (caused by increased levels of neutrophils and inflammatory cytokines) [29]. Therefore, invasive respiratory support is needed more frequently than noninvasive respiratory support to treat children with severe RSV infection [30] because invasive respiratory support provides an adequate positive end-expiratory pressure. Moreover, infants, particularly those aged under 5 months, are more likely to be hospitalized than children aged over 12 months [1, 30, 31]. This correlation may be related to the increase in the bronchiolar lumen with age. Pranlukast might decrease mucus

**Table 4. Relative likelihood of requiring respiratory support among infants treated with pranlukast according to the propensity score-matched analysis and the inverse probability of treatment-weighted analysis.**

| | Total sample size | Number treated with pranlukast | Number treated without pranlukast | Odds ratio (95% confidence interval) | P value |
|---|---|---|---|---|---|
| Propensity score matching | 294 | 147 | 147 | 0.27 | 0.01 |
| | | | | (0.08–0.79) | |
| Inverse probability of treatment weighting | 592 | 198 | 294 | 0.22 | 0.01 |
| | | | | (0.07–0.63) | |

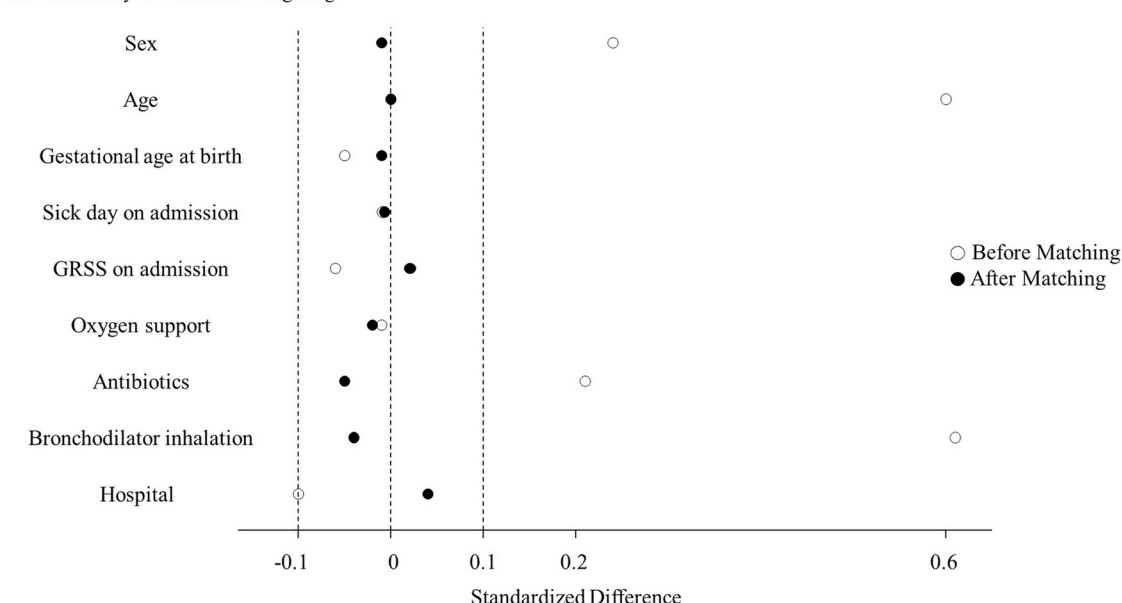

**Fig 4. Covariate balance for inverse probability of treatment weighting of infants treated with and without pranlukast.**

production and deposition of cellular debris by directly inhibiting neutrophils and inflammatory cytokines directly, and indirectly through dendritic cells and T cells [13–15, 32–34].

Regarding the length of hospital stay, the median length of hospital stay was not shortened by the use of pranlukast in this study or previous studies [22, 23]. However, the length of the hospital stay in infants who received respiratory support was significantly longer than that of infants who did not receive respiratory support (median [IQR]; 8 days [5–9 days] vs. 4 days [3–6 days]; $P$ <0.01) when the propensity score-matched infants were assigned to two groups depending on whether they received respiratory support. In this study, the small sample size of infants with respiratory support after matching might have contributed to the lack of an association between pranlukast treatment and the length of the hospital stay. Therefore, prospective studies with appropriate sample sizes are needed to assess whether there is an association between pranlukast treatment and the length of hospital stay.

In this study, similar to previous studies [21–23], treatment with pranlukast was not associated with an improved acute severity score among infants with RSV infection. Contrary to this finding, Zedan et al. [19] and Keskin et al. [20] found that montelukast improved the severity score of acute viral bronchiolitis in randomized controlled trials; however, they did not report the number of patients infected with RSV. A limitation of previous studies is that they evaluated the severity of RSV infection using different scoring systems in different studies and nonspecific RSV infection scoring systems. To address the lack of specificity, we used the GRSS to evaluate the severity of RSV infection. Using the GRSS, we could score the severity chronologically over the course of RSV infection [24]. Moreover, it provided a useful guide for decision-making regarding providing respiratory support for infants with RSV infection [35]. Although pranlukast did not improve the symptoms of acute RSV infection, it could help avoid the initiation of respiratory support. There are three possible reasons for the lack of an association between pranlukast and the GRSS. First, the sample size may have been too small to reveal an association. Second, the GRSS was calculated using the worst signs in the day because this study was retrospective. Last, the parameters of GRSS, presence of wheeze, rales/rhonchi,

retractions, cyanosis, lethargy, and poor air movement, were measured dichotomously; hence, the severity of these variables was not considered. For example, terminal expiratory wheezing (audible only with a stethoscope) and inspiratory and expiratory wheezing (audible without a stethoscope) were both assigned the same score.

This study has certain limitations. First, it was a retrospective study and not a double-blind, randomized controlled trial. Propensity score matching and inverse probability of treatment weighting were used to address the limitations of the study design and control for confounding. However, there is still a possibility of confounding by unmeasured covariates, such as cigarette smoke exposure and a family history of atopy. There might be information bias about starting respiratory support because this study was not double-blinded. However, the results confirm that pediatricians started respiratory support in patients with more severe respiratory dysfunction, as evidenced by the GRSS on starting respiratory support or at the time of the most severe point of the illness being significantly higher in infants who received respiratory support than in infants who did not receive respiratory support. In addition, there was no tendency to treat infants with a more severe disease with pranlukast, as evidenced by the lack of a significant difference in the GRSS according to the treatment of infants by pranlukast among infants who received respiratory support. Second, pranlukast was administered to infants with RSV infection because its use is covered by health insurance in Japan. Previous studies have assessed the effectiveness of montelukast [19–23]. However, pranlukast and montelukast have the same pharmacological effects [12, 36], and both block cysteinyl leukotriene receptor 1 [13, 14]. Therefore, the present study results are likely to be generalizable to other LTRAs. Third, the inclusion criteria of this study were infants aged less than 10 months without a high risk of exacerbation of RSV infection. However, the hospitalization rate of RSV infection is 2–5 times higher in infants aged <5 months than in those aged ≥5 months [1, 31]. In addition, more than 80% of hospitalized infants with RSV infection do not have any underlying conditions [4]. Therefore, the inclusion and exclusion criteria used in this study are not major limitations. Fourth, we did not examine whether the patients had coinfections with other viruses. To address these limitations, a multicenter, double-blind, randomized controlled trial should be conducted to confirm the effect of pranlukast treatment on the likelihood of requiring respiratory support.

In conclusion, the use of pranlukast was associated with a reduced likelihood of children requiring respiratory support. Pranlukast may be an effective primary treatment for infants with RSV infection, given the lack of availability of specific treatment. Prospective studies are required to confirm the beneficial effect of pranlukast treatment in infants with RSV infection.

## Supporting information

**S1 Dataset.**
(XLSX)

## Author Contributions

**Conceptualization:** Jun Kubota, Sho Takahashi.

**Data curation:** Jun Kubota.

**Formal analysis:** Jun Kubota, Sho Takahashi.

**Investigation:** Jun Kubota, Takayuki Suzuki, Akira Ito, Naoe Akiyama, Noriko Takahata.

**Methodology:** Jun Kubota, Sho Takahashi.

**Project administration:** Jun Kubota.

**Supervision:** Jun Kubota, Sho Takahashi.

**Validation:** Jun Kubota.

**Visualization:** Jun Kubota.

**Writing – original draft:** Jun Kubota.

**Writing – review & editing:** Jun Kubota, Sho Takahashi, Takayuki Suzuki, Akira Ito, Naoe Akiyama, Noriko Takahata.

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
