## [Decision Letter · Decision Letter 0]

6 May 2022

PONE-D-22-06884Pranlukast treatment and the use of respiratory support in infants with respiratory syncytial virus infectionPLOS ONE

Dear Dr. Kubota,

Thank you for submitting your manuscript to PLOS ONE. After careful consideration, we feel that it has merit but does not fully meet PLOS ONE’s publication criteria as it currently stands. Therefore, we invite you to submit a revised version of the manuscript that addresses the points raised during the review process.

We look forward to receiving your revised manuscript.

Kind regards,

Nades Palaniyar, MSc., PhD.

Academic Editor

PLOS ONE

Journal Requirements:

Reviewers' comments:

Reviewer's Responses to Questions

**Comments to the Author**

1. Is the manuscript technically sound, and do the data support the conclusions?

Reviewer #1: Yes

Reviewer #2: Yes

2. Has the statistical analysis been performed appropriately and rigorously? 

Reviewer #1: Yes

Reviewer #2: Yes

3. Have the authors made all data underlying the findings in their manuscript fully available?

Reviewer #1: Yes

Reviewer #2: Yes

4. Is the manuscript presented in an intelligible fashion and written in standard English?

Reviewer #1: Yes

Reviewer #2: Yes

5. Review Comments to the Author

Reviewer #1: Kubota et al describe findings from a group of infants admitted with acute RSV infection. They demonstrate that infants treated with a leukotriene receptor antagonist required less respiratory support than matched infants who were not treated. Of note, symptoms and length of stay were not impacted. This is a logical and easy read to paper. I do have a few comments/questions for the authors.

1) In the abstract, without reading the paper it is unclear if the odds ratio and length of stay data are derived from the matched group of infants of the full dataset of infants (n=492). Is there a way to make it clearer that the only analysis performed was on the matched infants?

2) Page 4, line 52. A vaccine is not really a specific treatment and so perhaps you can remove this word from the sentence the way it is currently written?

3) It is commendable that the GRSS was derived from retrospective data. Did the authors have difficulties with missing data?

4) Line 278 mentions smoking and atopy as potential confounders...were these data not captured in the charts?

5) Line 300 says: "the use of pranlukast reduced the likelihood of children requiring respiratory support..."

I would suggest changing to "the use of pranlukast was associated with a reduced likelihood of children requiring respiratory support." (ie less causal language)

Reviewer #2: The paper is interesting and well written. Below are minor questions that can help improve the paper:

Was there any analysis carried out to determine if assigning patients who began pranlukast treatment the same day as respiratory support in the “no pralukast” group affected the results?

Was there a sex difference observed?

Was background and race effect studied?

What would the overall advice be to a clinician when treating these patients and under what parameters would you recommend the use of prankulast?

6. PLOS authors have the option to publish the peer review history of their article (what does this mean?). If published, this will include your full peer review and any attached files.

Reviewer #1: **Yes: **Theo J Moraes

Reviewer #2: No

---

## [Author Response · Author response to Decision Letter 0]

10 May 2022

Response to Reviewer 1

1) In the abstract, without reading the paper it is unclear if the odds ratio and length of stay data are derived from the matched group of infants of the full dataset of infants (n=492). Is there a way to make it clearer that the only analysis performed was on the matched infants?

Response:

Thank you for your comment. As per your suggestion, we have revised the text of our manuscript as follows:

“In the propensity score-matched analysis, pranlukast use was associated with a significantly lower chance of needing respiratory support (odds ratio: 0.27, 95% confidence interval: 0.08–0.79; P =0.01); however, the length of hospital stay (median: 4 days) and the GRSS (median: 2.804 and 2.869 for infants treated with and without pranlukast, respectively) did not differ significantly between propensity score-matched pairs.” (Abstract, Results, Lines 33–38)

2) Page 4, line 52. A vaccine is not really a specific treatment and so perhaps you can remove this word from the sentence the way it is currently written?

Response:

We agree with your suggestion. We have removed the words and revised the text of our manuscript as follows:

“Supportive care is the only primary treatment because there is no specific treatment for RSV [5–7].” (Introduction, Lines 46–47)

3) It is commendable that the GRSS was derived from retrospective data. Did the authors have difficulties with missing data?

Response:

We thank you for your compliment. We had some difficulties with missing data and excluded 12 patients because of incomplete medical records.

4) Line 278 mentions smoking and atopy as potential confounders...were these data not captured in the charts?

Response:

No, these data were not captured in the charts. Therefore, we have mentioned that there is still a possibility of confounding by unmeasured covariates as a limitation.

5) Line 300 says: "the use of pranlukast reduced the likelihood of children requiring respiratory support..."

I would suggest changing to "the use of pranlukast was associated with a reduced likelihood of children requiring respiratory support." (ie less causal language)

Response:

Thank you for your comment. As per your suggestion, we have revised the text of our manuscript as follows:

“In conclusion, the use of pranlukast was associated with a reduced likelihood of children requiring respiratory support.” (Discussion, Lines 314–315)

 

Response to Reviewer 2

Was there any analysis carried out to determine if assigning patients who began pranlukast treatment the same day as respiratory support in the “no pralukast” group affected the results?

Response:

Thank you for your comment. We did not conduct an analysis as you suggested because we did not have data on the time lag from administration of pranlukast to starting respiratory support. Therefore, we categorized these patients into no pranlukast group.

Was there a sex difference observed?

Response:

Before propensity score-matching, male infants were significantly more likely to be treated with pranlukast than female infants (p < 0.01). After propensity-score matching, there no difference between males and females. We have indicated this in Table 1.

Was background and race effect studied?

Response:

Thank you for your comment. A previous study reported that black infants were less severe RSV bronchiolitis than white those [Bradley JP, et al. Severity of respiratory syncytial virus bronchiolitis is affected by cigarette smoke exposure and atopy. Pediatrics 2005; 115:e7–14.]. However, almost all patients in this study were Japanese. Therefore, we have not mentioned it in the limitation section.

What would the overall advice be to a clinician when treating these patients and under what parameters would you recommend the use of pranlukast?

Response:

Thank you for your question. Based on the results of this study, we recommend that pediatricians should start to treat infants with RSV infections with pranlukast as soon as they are diagnosed because there is no specific treatment for RSV and there were no side effects of pranlukast observed in this study. However, as this study was a retrospective study, so we hesitate to make strong recommendations. Therefore, we have revised the manuscript as follows:

“In conclusion, the use of pranlukast was associated with a reduced the likelihood of children requiring respiratory support. Pranlukast may be an effective primary treatment for infants with RSV infection, given the lack of availability of specific treatment. Prospective studies are required to confirm the beneficial effect of pranlukast treatment in infants with RSV infection.” (Discussion, Lines 314–317) 

Journal Requirements:

Response:

We have checked that the manuscript meets the PLOS ONE style requirements and that the reference list is complete and correct.

---

## [Editor Report · Decision Letter 1]

13 May 2022

Pranlukast treatment and the use of respiratory support in infants with respiratory syncytial virus infection

PONE-D-22-06884R1

Dear Dr. Kubota,

We’re pleased to inform you that your manuscript has been judged scientifically suitable for publication and will be formally accepted for publication once it meets all outstanding technical requirements.

Kind regards,

Nades Palaniyar, MSc., PhD.

Academic Editor

PLOS ONE

Additional Editor Comments (optional): This a very useful study. Well done.
---

## [Editor Report · Acceptance letter]

19 May 2022

PONE-D-22-06884R1 

Pranlukast treatment and the use of respiratory support in infants with respiratory syncytial virus infection 

Dear Dr. Kubota:

I'm pleased to inform you that your manuscript has been deemed suitable for publication in PLOS ONE. Congratulations! Your manuscript is now with our production department. 

Kind regards, 

on behalf of

Dr. Nades Palaniyar 

Academic Editor

PLOS ONE